# Enhanced interfacial water dissociation on a hydrated iron porphyrin single-atom catalyst in graphene

Laura Scalfi [1,3], Maximilian R. Becker [1,3], Roland R. Netz [1] & Marie-Laure Bocquet [2✉]

Single Atom Catalysis (SAC) is an expanding field of heterogeneous catalysis in which single metallic atoms embedded in different materials catalyze a chemical reaction, but these new catalytic materials still lack fundamental understanding when used in electrochemical environments. Recent characterizations of non-noble metals like Fe deposited on N-doped graphitic materials have evidenced two types of $Fe-N_4$ fourfold coordination, either of pyridine type or of porphyrin type. Here, we study these defects embedded in a graphene sheet and immersed in an explicit aqueous medium at the quantum level. While the Fe-pyridine SAC model is clear cut and widely studied, it is not the case for the Fe-porphyrin SAC that remains ill-defined, because of the necessary embedding of odd-membered rings in graphene. We first propose an atomistic model for the Fe-porphyrin SAC. Using spin-polarized ab initio molecular dynamics, we show that both Fe SACs spontaneously adsorb two interfacial water molecules from the solvent on opposite sides. Interestingly, we unveil a different catalytic reactivity of the two hydrated SAC motives: while the Fe-porphyrin defect eventually dissociates an adsorbed water molecule under a moderate external electric field, the Fe-pyridine defect does not convey water dissociation.

[1] Fachbereich Physik, Freie Universität Berlin, Arnimallee 14, 14195 Berlin, Germany. [2] Laboratoire de Physique de l'École Normale Supérieure, ENS, Université PSL, CNRS, Sorbonne Université, Université Paris Cité, F-75005 Paris, France. [3] These authors contributed equally: Laura Scalfi, Maximilian R. Becker. ✉email: marie-laure.bocquet@ens.fr

Since two decades, the field of heterogeneous catalysis has seen a surge of interest in single-atom catalysts (SAC) comprised of single transition metal atom active sites[1,2]. The original experiments were performed on oxide supports, with the insertion of individual Au or Pt atoms into cerium oxide for the water-gas shift reaction[3] and later with single Pt atoms inserted into an iron oxide for CO oxidation[4]. Single-atom catalysts quickly invested the active field of electrocatalysis, where these diluted catalysts were immersed into liquid electrolytes. The SAC now even extends to the field of bionanocatalysis with promising anti-bacterial results[5]. In search of scalability, individual noble metals like Pt and non-noble metals like Co and Fe have been successfully inserted into various carbonaceous supports, such as porous graphene[6], graphene oxide[7] and also other 2D materials like $MoS_2$[8].

A practical question that arises is how to best graft single metals on 2D materials, while avoiding metal sintering or clustering. Several synthesis strategies have successfully been implemented: the most common technique consists of first pre-doping graphitic materials like graphene or graphene oxide with nitrogen atoms via heating under $NH_3$ atmosphere. Upon addition of the metal as a salt and subsequent pyrolysis, single metal ions coordinate in the vacancies forming four strong metal–N bonds[7,9,10]. Two other synthesis strategies for SACs include high-energy ball-milling of the metal phthalocyanine molecule with graphene nanosheets[11] and a surfactant-assisted synthesis method using a mixture of metal salt, a graphitic carbon nitride and a polymer surfactant[12,13].

With the aid of XANES (X-ray Absorption Near Edge Structure) spectroscopy and density functional theory (DFT) modeling[6,9] or X-ray photoelectron spectroscopy[7,12], two possible SAC grafting geometries around the metal core were proposed: one of pyridine type, with the metal coordinated by 4 pyridinic ligands (six-membered rings), and one of porphyrin type, with the metal coordinated by 4 pyrroles ligands (five-membered rings). Both defects have been found to coexist in these SACs, and recently Zhang et al.[14] have successfully converted the iron pyridine defects into the iron porphyrin ones after a $NH_3$-assisted pyrolysis. These fourfold motives, labeled M-$N_4$ SACs, have been widely tested experimentally in a large range of electrochemical reactions, in particular for the oxygen reduction reaction (ORR), and using different metal centers, where Fe was found to be a promising catalytically active non-noble metal[15]. Interestingly, a recent study compared the two defects mentioned above and found a different predicted reactivity[16]. Indeed, by a combined experimental and theoretical study, the Co SAC porphyrin motif (also called pyrrole in the literature) was found most efficient for the 2 electrons ORR pathway, leading to $H_2O_2$ formation, while the Co SAC pyridine motif works better for the 4 electrons ORR, leading to $H_2O$ production. Note that the latter selectivity for ORR drastically depends on the metal atom: for example, high-purity Fe SAC porphyrin defects in proton-exchange membrane fuel cells rather exhibit complete four electrons reaction selectivity[14].

From a theoretical point of view, these SACs are easily modeled at the quantum level either as molecular models or as small periodic models, allowing for the quantitative prediction of electrochemical pathways in vacuum adding electrons and protons (including several studies with Fe[17,18]). However, modeling an explicit solution environment around solids plays an important role and has recently revealed many unexpected interfacial reactivities like charge adsorption on 2D materials[19] and enhanced water dissociation on defected[20] or hybridized 2D materials[21], or in nanoconfined water[22,23]. Regarding single-atom catalysis, Patel et al.[24] recently suggested by comparing a Cu-modified covalent triazine framework catalyst with a Cu(111)

surface that solvation effects are probably larger for SACs than for conventional metal catalysts. Up to now however, only a single theoretical study considered the Fe-pyridine defect embedded in a graphene sheet and immersed in an active chemical environment, i.e. including the solvent water molecules interfacing the SAC, using extensive DFT molecular dynamics (DFT-MD)[25]. That study also emphasized the crucial need for taking into account explicit solvation in order to represent correctly the dynamical reactive interfacial processes at stake.

On a technical level, static DFT studies[17,26] highlighted the importance of performing quantum calculations with care and using spin-polarized simulations as soon as dealing with magnetic atoms like Fe or Co. Another important aspect often poorly discussed is the full atomistic model in which the SAC is embedded. In previous studies, the simulation box was only slightly larger than the catalytic center itself. However, this imposes large constraints on the geometry of the SAC defects and is not a realistic representation of the graphene support. Indeed, the reported experimental density of these defects is between 0.5 and 4 wt%[27]. Therefore, we argue here that larger unit cells should be used to ensure a better representation of the systems and that the local defect created by the SAC insertion induces only small local perturbations[26].

In this theoretical work, we investigate iron-based SAC embedded in graphene, motivated by the low cost of the materials. We compare the reactivity of two Fe–$N_4$ SAC defects in operando, i.e. including explicit liquid water by means of extensive spin-polarized DFT-MD simulations and employing large periodic unit cells (corresponding to a density of defects of 2.8–2.9 wt%) to mitigate the induced strain in graphene. As a result, we demonstrate that, while two water molecules of the solvent readily coordinate in an axial fashion the Fe atom on both defects, the Fe-porphyrin defect has a significant stabilizing effect on an adsorbed hydroxide ion. By applying a moderate electric field perpendicular to the graphene support, the Fe-porphyrin defect leads to the deprotonation of one of the adsorbed water molecules. In contrast, the Fe-pyridine defect remains inactive under the same external stimulus; the adsorbed water molecules stay coordinated with the Fe metal ion, effectively passivating or protecting the Fe core. This shows that Fe-porphyrin is better suited for water protolytic applications than Fe-pyridine.

## Results and discussion

**Geometries and electronic structure of bare SAC defects in gas phase.** As mentioned in the introduction, two configurations for embedding Fe in N-doped graphene were proposed in earlier studies[6,9,12,17]. The structure most widely studied, which we denote Fe-pyridine defect, is pictured in Fig. 1a: two consecutive carbon atoms forming a vertical bond are removed (not shown) and this di-vacancy is filled by one metal cation, while the four edge carbon atoms of the as-formed di-vacancy are substituted by nitrogen atoms. A second configuration has been proposed by periodically replicating a porphyrin macrocycle, leading to a compensation of each pyrrole group by one eight-member ring placed in anti-position along the diagonal directions[16,17,28]. Here, we propose an alternative way to embed the Fe-porphyrin defect: as shown in Fig. 1b (see also Supplementary Fig. 1), we incorporate a porphyrin into a large graphene sheet (approximately 20 by 20 Å) and solve the non-commensurability of porphyrin with the graphene structure by inserting two seven-member rings and a five-member ring on the left and right sides of the porphyrin, reminiscent of a Stone–Wales defect[29], highlighted in yellow on Fig. 1b. We have checked that the induced strain in graphene remains small and localized to the defect. Indeed, after geometry-

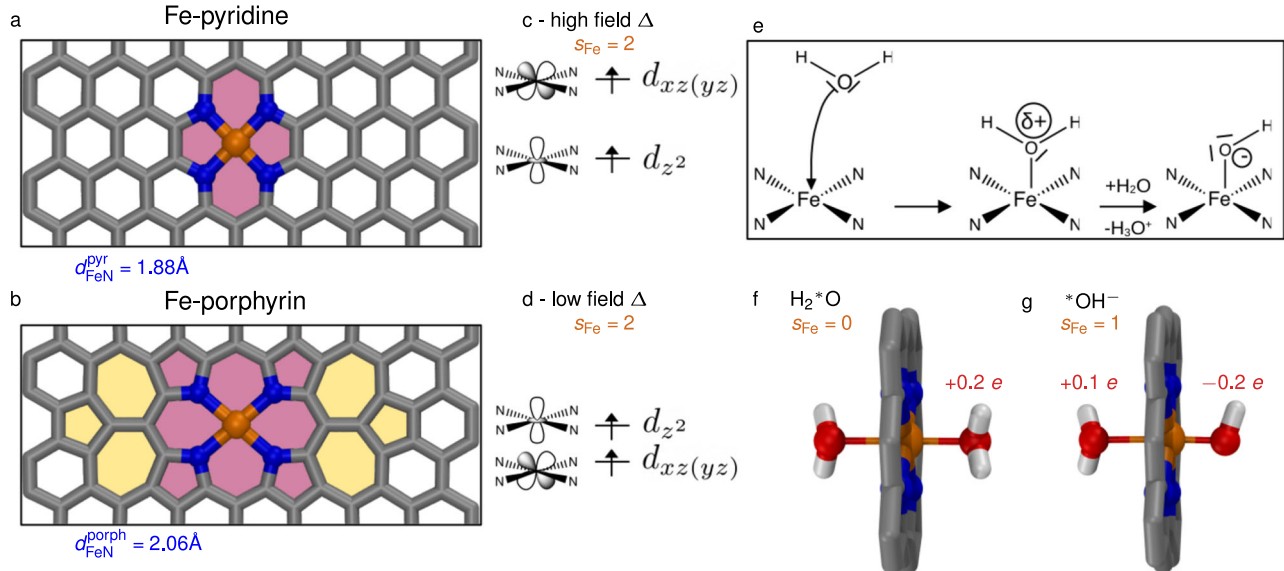

**Fig. 1 Bare defects in gas-phase.** Top views of the DFT-optimized Fe-pyridine (**a**) and Fe-porphyrin defects (**b**) embedded in graphene. The simulation box is cropped in the vertical direction: for snapshots of the entire unit cell, see Supplementary Fig. 1. The colors are as follows: C—gray, Fe—orange, N—blue, O—red, H—white. Pink areas highlight the defect pattern and the yellow areas show the graphene reconstruction with seven- and five-membered rings. (**c, d**) Representation of the magnetic *d*-orbitals of the Fe atom in SAC extracted from spin-polarized projected densities of states (pDOS, explicitly shown in Supplementary Fig. 3), together with corresponding Fe spin polarizations $s_{Fe}$. Water as a ligand: (**e**) Mechanism of a single water molecule adsorption on the Fe SAC defect, where the Fe atom acts as a Lewis acid followed by auto-dissociation of the adsorbed water. Close-up DFT structures of the Fe SAC coordinated by either two water molecules (**f**) or a water molecule and a hydroxide ion (**g**), along with computed total charges of the adsorbed $H_2$*O or *$OH^-$ and Fe spin polarizations $s_{Fe}$. The full box is shown in Supplementary Fig. 1. Both defects yield similar charge and spin values, for more details see Table 2.

optimization at the DFT level, the obtained C–C bonds are only slightly longer than the average bond length for the pristine graphene $d_{CC}^{graphene} = 1.42$ Å: the maximal C–C length is $d_{CC}^{max} = 1.5$ Å in the five-member rings only and the full bond distribution is given in Supplementary Fig. 2. This allows our Fe-porphyrin structure to stay stable throughout all further simulations and to maintain a planar catalytic site. The main geometrical difference between the two SAC structures resides in the Fe-N distance: $d_{FeN}^{pyr} = 1.88$ Å in the Fe-pyridine defect is significantly smaller than $d_{FeN}^{porph} = 2.06$ Å for the Fe-porphyrin defect, in fine agreement with recent EXAFS (Extended X-ray Absorption Fine Structure) measurements made for related cobalt SAC defects[16].

Because the Fe cation center can be magnetic, we perform static DFT and DFT-MD simulations with the simulation package CP2K[30] on the PBE level[31,32] using unrestricted (spin-polarized) Kohn Sham equations (UKS) with imposed spin multiplicity *M* of the total system. Simulation details are given in the Methods section and in the Supplementary Methods.

Because of the localized *d*-orbitals on the isolated Fe metal ion, the delocalization error of GGA (generalized gradient approximation) functionals might lead to inaccurate results, while hybrid functionals such as PBE0[33] correct for this and are therefore more reliable for predicting properties of the transition metal ions present in SACs[17,24,26]. However, this is prohibitively computationally expensive for running MD simulations. To check for the severity of this approximation, we provide a comparison of the adsorption energies of single molecules onto the defects and of the energy difference between spin states calculated with PBE and PBE0. The data is given in Supplementary Figs. 5, 6 and discussed in the sections "Single adsorbed waters in gas phase" and "SAC in liquid water".

We investigate the impact of spin settings in our DFT simulations by comparing results for two different spin multiplicities of the whole system—UKS *M* = 1 and UKS *M* = 3. As

**Table 1 Bare Fe SAC defects in gas phase.**

| Spin Settings | E (kcal mol⁻¹) | $s_{Fe}$ | $q_{Fe}$ [e] |
|---|---|---|---|
| **Fe-pyridine** | | | |
| UKS $M = 1$ | −655501 | 0.0 | 0.18 |
| UKS $M = 3$ | −655512 | 2.11 | 0.22 |
| **Fe-porphyrin** | | | |
| UKS $M = 1$ | −640978 | 2.07 | 0.46 |
| UKS $M = 3$ | −640984 | 2.25 | 0.49 |

Potential energy *E*, Fe spin polarization $s_{Fe}$ and Fe charges $q_{Fe}$ for different spin settings (UKS $M = 1$ and UKS $M = 3$). See Methods section and Supplementary Methods for more details.

expected, the spin polarization (taken as the number of unpaired electrons) mostly localizes on the Fe atom[17]. In Table 1, we report extracted charges $q_{Fe}$ and spin polarizations $s_{Fe}$ on the Fe atom for different settings. For Fe-pyridine, both a low and a high spin state can be found, with $s_{Fe} = 0$ (no unpaired electrons localized on Fe) and $s_{Fe} \approx 2$ (2 unpaired electrons localized on Fe) respectively. For Fe-porphyrin, we observe the high spin state with $s_{Fe} \approx 2$ for both UKS settings (*M* = 1 and *M* = 3). In both bare defects, the high spin state (corresponding to a Fe triplet spin state, which does not necessarily correspond to the total system multiplicity *M*) is energetically favorable by about 10-20 kcal mol⁻¹. We also observe that the high spin states have higher charges on the iron atom and the Fe-porphyrin defect shows slightly higher charge values with respect to Fe-pyridine. Interestingly, a combined experimental and theoretical study has raised similar conclusions stating that the valence state of Fe in the Fe-porphyrin defect is more positive than that in the Fe-pyridine, pointing to a higher chemical reactivity of the Fe-porphyrin defect[14]. This demonstrates the importance of using spin-polarized simulations allowing for non-zero spin on the Fe core for bare defects in vacuum.

We further analyze spin-polarized densities of states projected on atomic orbitals of the Fe atom (pDOS, schematized in Fig. 1c, d and shown in Supplementary Fig. 3). The Fe-porphyrin defect pDOS is very similar to the pDOS of the molecular porphyrin, except that the $d_{xz}$ and $d_{yz}$ degeneracy is slightly lifted due to the rectangular distortion of the Fe environment. For both defects, we find the following occupation of the $d$-orbitals: $d_{xy}^{(2)}[d_{xz}, d_{yz}]^{(3)}d_{z^2}^{(1)}$ (where we cannot assign the $d_{xz}$ and $d_{yz}$ separately), which is consistent with ligand field theory and previous calculations for porphyrins[34,35]. The Fe $d_{z^2}$ and the $d_{xz(yz)}$ orbitals (drawn in Fig. 1c, d) are therefore magnetic orbitals i.e containing one unpaired electron each. In line with the smaller Fe–N distance $d_{FeN}$ in Fe-pyridine, we measure a higher ligand field splitting in Fe-pyridine: the two orbitals lie further apart and there is a larger destablization of the $d_{xz(yz)}$ magnetic orbital. In contrast, the lengthening of Fe–N bond in the Fe-porphyrin defect leads to a smaller ligand field destablization of the $d_{xz(yz)}$ orbital, placing it close to but below the $d_{z^2}$ orbital, leading to an energetic inversion between the two magnetic orbitals.

**Single adsorbed waters in gas-phase**. To investigate the reactivity of these Fe SAC defects in aqueous conditions, we first report static DFT results for single water molecules adsorbed on the catalytic site in gas phase. From a chemical point of view, the electron-deficient (positively charged) Fe core of SAC defects should ligate water spontaneously via a nucleophilic addition of the lone pair of the water oxygen, as drawn in Fig. 1e, and eventually induce the adsorbed water to auto-dissociate, loose one proton and form an adsorbed hydroxide anion.

We explore two different situations: the adsorption of two water molecules on either side of the catalytic site, as in Fig. 1f and noted $H_2{}^*O$, and the adsorption of a water molecule and a hydroxide ion, as in Fig. 1g, noted $^*OH^-$. (Here and in the following, $^*O$ denotes oxygen atoms adsorbed on the Fe metal ion). In the case of $^*OH^-$, a single proton is added far from the hydroxide ion to ensure the neutrality of the simulation cell. Note that to ensure a $+1$ charge on the proton, no basis set is attributed to the H atom. We prepare different initial configurations and optimize their ground-state geometry at the PBE-DFT level. The results for these complexes are similar and the values for the lowest energy configuration are shown in Table 2 for different spin settings. In the $H_2{}^*O$ configuration, we find distinct low and high spin states for both defects, with a slightly lower energy reported for the high spin states ($\leq 5$ kcal mol$^{-1}$). For each system, we compute the adsorption energies of the water molecules and/or the hydroxide ion onto the Fe-defect. The adsorption energy of two water molecules is about $-20$ kcal

mol$^{-1}$, while we observe a very strong adsorption of $^*OH^-$ onto both defects with an adsorption energy between $-110$ to $-135$ kcal mol$^{-1}$. Interestingly, $\Delta E_{ads}$ is globally more negative for Fe-porphyrin configurations than for Fe-pyridine ones, i.e. water molecules and the hydroxide ion adsorb more strongly on Fe-porphyrin than on Fe-pyridine. Values obtained from PBE0, given in Supplementary Fig. 5, are in qualitative agreement with the conclusions above. The relative adsorption energies between Fe-pyridine and Fe-porphyrin are similar, nevertheless we find that $\Delta E_{ads}$ is more negative for PBE, with differences of about 2.5 kcal mol$^{-1}$ for absorption of water molecules and 10 kcal mol$^{-1}$ for adsorption of $^*OH^-$.

Among the different optimized systems, we observe different adsorption geometries on the catalytic site. For 2 adsorbed water molecules, we note that in the low spin states the distance between Fe and the adsorbed oxygen atom $^*O$ $d_{Fe^*O} \approx 2.06$ Å is smaller than for high spin states, where it equals $2.3-2.4$ Å. Additionally, we observe that water molecules carry a positive partial charge, with a total charge on each water molecule of about $0.2$ $e$ in the low spin states and $0.1$ $e$ in the high spin states. This indicates a net electron transfer from the two water molecules to the graphene sheet of $0.4$ $e$ in the low spin state, and suggests a Lewis acid-type mechanism for the Fe–$^*OH_2$ interaction, as pictured in Fig. 1e. We also note that high spin states have a weaker interaction with water, as suggested by the larger Fe–$^*O$ distance and the smaller charge transfer. For the $^*OH^-$ configuration, we observe an intermediary spin state with $s_{Fe} \approx 0.9-1.0$ (with a slightly higher value for Fe-pyridine). Upon water deprotonation, the Fe–$^*O$ distance of the hydroxide oxygen shortens to $d_{Fe^*OH^-} = 1.8$ Å. At the same time, the Fe–$^*O$ distance of the opposite water molecule lengthens up to $d_{Fe^*OH_2}^{pyr} = 2.35$ Å for Fe-pyridine and $d_{Fe^*OH_2}^{porph} = 2.2$ Å for Fe-porphyrin, leading to a reduced positive charge on the water molecule with respect to the previous calculations, with a charge transfer of $0.1-0.15$ $e$. Such structural asymmetry between axial ligands is a direct signature of the trans-effect well-known in coordination chemistry[36]. The charge on the hydroxide anion is between $-0.2$ and $-0.3$ $e$, so a large portion of the additional negative charge is transferred to the graphene substrate.

Therefore, we observe that the spin state of the Fe core evolves with water chemisorption and water dissociation, which confirms the importance of using spin-polarized DFT simulations to model these gas-phase systems. Both $M = 1$ and $M = 3$ system spin multiplicities give similar results for the $^*OH^-$ case, leading to an intermediary spin state. In the $H_2{}^*O$ case in contrast, the setting $M = 3$ leads to a weaker interaction with water molecules. This is consistent with PBE0 results shown in Supplementary Note 3.

**Table 2 Water and hydroxide adsorption on Fe SAC defects in gas phase.**

| | $H_2{}^*O$ | | | | | $H_2{}^*O_L$ and $^*O_RH^-$ | | | | | | |
|---|---|---|---|---|---|---|---|---|---|---|---|---|
| **Spin Settings** | $\Delta E_{ads}$ [kcal mol$^{-1}$] | $s_{Fe}$ | $q_{Fe}$ [e] | $q_{H_2{}^*O}$ [e] | $d_{Fe^*O}$ [Å] | $\Delta E_{ads}$ [kcal mol$^{-1}$] | $s_{Fe}$ | $q_{Fe}$ [e] | $q_{H_2{}^*O_L}$ [e] | $q_{^*O_RH^-}$ [e] | $d_{Fe^*O_L}$ [Å] | $d_{Fe^*O_R}$ [Å] |
| **Fe-pyridine** | | | | | | | | | | | | |
| UKS $M = 1$ | −22.2 | 0.0 | 0.12 | 0.20 | 2.06 | −122.3 | 1.0 | 0.21 | 0.11 | −0.20 | 2.38 | 1.81 |
| UKS $M = 3$ | −14.2 | 2.1 | 0.24 | 0.08 | 2.50 | −110.9 | 0.9 | 0.20 | 0.12 | −0.20 | 2.32 | 1.81 |
| **Fe-porphyrin** | | | | | | | | | | | | |
| UKS $M = 1$ | −27.6 | 0.0 | 0.40 | 0.20 | 2.07 | −135.6 | 0.9 | 0.44 | 0.15 | −0.19 | 2.18 | 1.80 |
| UKS $M = 3$ | −22.3 | 2.4 | 0.52 | 0.11 | 2.32 | −124.8 | 0.9 | 0.44 | 0.15 | −0.17 | 2.18 | 1.79 |

BSSE-corrected adsorption energy $\Delta E_{ads}$ of the water molecules and/or hydroxide ion onto the Fe-defects, Fe spin polarization $s_{Fe}$, Fe charges $q_{Fe}$, total charges on the adsorbed $H_2{}^*O$ and $^*OH^-$ and Fe–$^*O$ distances for different spin settings (UKS $M = 1$ and UKS $M = 3$). Data is shown for DFT-optimized configurations coordinated by either two water molecules or a water molecule and a hydroxide ion in gas phase, corresponding to the structures given in Fig. 1f, g. See Methods section and Supplementary Methods for more details.

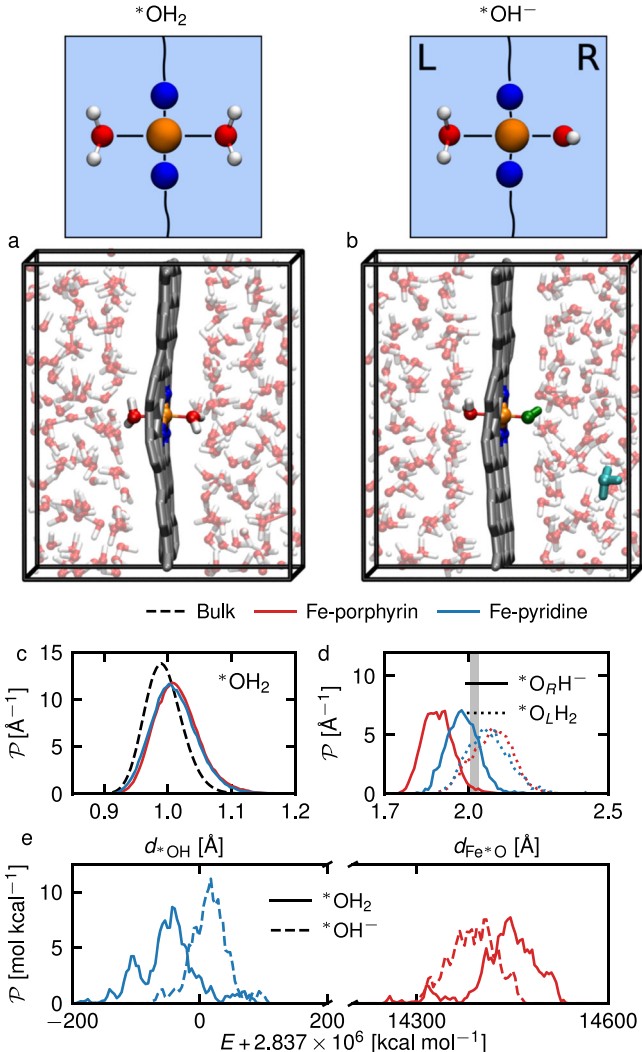

**Fig. 2 Characterization of the fully hydrated Fe-defects.** Schematic representations and snapshots of the simulation boxes with two adsorbed *OH₂ (**a**) or with one adsorbed *OH₂ on the left side, one adsorbed *OH⁻ on the right side and a hydronium ion far away from the catalytic site (**b**). The colors are as follows: C—gray, Fe—orange, N—blue, O—red, H—white, OH⁻—dark green, H₃O⁺—cyan. We show results for the Fe-porphyrin (red lines) and the Fe-pyridine (blue lines) hydrated defects. **c** *O–H distance $d_{*OH}$ distribution for simulations with two adsorbed water molecules (case **a**), averaged over the two oxygens atoms *O adsorbed to the Fe atom, compared to the O–H distance distribution for water molecules in bulk (black dashed line). **d** Fe-*O distance $d_{Fe*O}$ distributions from simulations with initially a single hydroxide ion adsorbed on the right side of the catalytic site (case **b**), for the right adsorbed oxygen (solid lines) and for the left adsorbed oxygen (dotted lines). The vertical gray line shows the average distance for the simulations with two adsorbed water molecules (case **a**). In the case of Fe-pyridine, the distribution was computed only on configurations before reprotonation of the hydroxide ion. **e** Total energy distributions for simulations with two adsorbed water molecules (case **a** - solid lines) and for simulations with initially a single hydroxide ion adsorbed on the catalytic site (case **b**—dashed lines).

**SAC in liquid water**. In the following, we investigate the activity of the Fe-embedded graphene structures in liquid aqueous conditions, using extensive DFT-MD simulations of the hydrated system shown in Fig. 2a and Supplementary Movies 1–2, where the simulation box containing the Fe SAC defects is filled with liquid water. Although we are aware that in many experiments the doped graphene sheet is often exposed to water on one side and to solid support on the other side, we choose this model because this configuration is frequently adopted in the literature and is also computationally advantageous to simulate.

For completeness, we run both UKS simulations with a system multiplicity $M = 1$ and $M = 3$. For a selection of configurations along these DFT-MD trajectories, we calculate the difference between these two settings and found the $M = 1$ setting to be the lowest in energy in the vast majority of the cases (see Supplementary Fig. 6, where we also give a comparison with PBE0 results). Therefore, in the following, we only show the results of DFT-MD simulations using a system multiplicity $M = 1$. Other spin settings in DFT-MD runs are analyzed in Supplementary Table 1 and Supplementary Fig. 4. During the equilibration of the DFT-MD simulations, we observe that two water molecules from both sides of the liquid slab readily adsorb onto the Fe metal ion at a distance of approximately 2 Å, in agreement with the previous section. In the following, we note the related adsorbed oxygens by *O to differentiate them from the rest of the water solvent molecules. During the production runs, we do not observe any exchange of these adsorbed water molecules on the timescale of our simulations (~10−20 ps—details are given in Table 3). Figure 2c shows the distribution of the *O–H bond length for the adsorbed water molecules for both defects (solid red line for the Fe-porphyrin and solid blue line for the Fe-pyridine), compared to the O–H bond length distribution in the bulk region far from the graphene sheet (dashed black line). We observe a significant lengthening of the *O–H bonds of the adsorbed water molecules with respect to the bulk value, which hints at a weakening of the *O–H bond and a decrease of the free energy barrier to water dissociation at the defect. It is interesting to note that no *O–H bond lengthening was observed in the gas phase calculations without surrounding solvent, depicted in Fig. 1e, which suggests that the hydrogen bonding in liquid water is crucial to facilitate water dissociation via proton transfer.

Quantifying the free energy barrier of deprotonation in the adsorbed state is however not possible because no spontaneous deprotonation occurs during the limited length of our unbiased simulations. Thus, we examine the stability of hydroxide in aqueous conditions, i.e. of the product of this deprotonation process. We prepare configurations with a hydroxide ion adsorbed on the right side of the catalytic site ($z > 0$), and a hydronium ion to ensure charge neutrality, placed initially as far as possible in the liquid water from the hydroxide, as shown in Fig. 2b. Figure 2d shows the distribution of the Fe–*O distance; the average value in the *OH₂ case is indicated by the vertical gray line (the distribution is identical for both defects). In the *OH⁻ case, the Fe–*O distance on the hydroxide side (shown by the solid lines) is shorter, similar to what we observed previously in single molecule calculations, while for the opposite adsorbed water it increases slightly (dotted lines). This trans-effect is larger for Fe-porphyrin (red lines) than Fe-pyridine (blue lines), indicating a stronger Fe–*OH⁻ bond in the Fe-porphyrin case. Figure 2e shows the total energy distributions along simulations with two adsorbed water molecules *OH₂ (solid lines) and with initially an adsorbed hydroxide anion *OH⁻ (dashed lines). Upon water deprotonation we find a decrease in average energy for Fe-porphyrin but not for Fe-pyridine. Although the distributions significantly overlap, the average total energy of the *OH⁻ simulations for Fe-pyridine is about 60 kcal mol⁻¹ higher than in the pure water simulations, while it is approximately 50 kcal mol⁻¹ lower in the case of Fe-porphyrin, as confirmed in Table 3.

Other characteristics of these MD simulations are reported in Table 3. In particular, we find spin states consistent with the

**Table 3 Fully hydrated Fe SAC defects.**

| Initial setup | Time [ps] | $\langle s_{Fe} \rangle$ | $\langle q_{Fe} \rangle$ [e] |
|---|---|---|---|
| **Fe-pyridine** | | | |
| *OH₂ | 17.3 | 0.0 | 0.16 |
| *OH⁻ | 11.4 | 0.05 | 0.17 |
| E-field *OH₂ | 16.9 | 0.0 | 0.17 |
| **Fe-porphyrin** | | | |
| *OH₂ | 20.8 | 0.4 | 0.44 |
| *OH⁻ | 18.1 | 0.8 | 0.46 |
| E-field *OH₂ | 20.5 + 16.9 | 0.5 & 0.9 | 0.45 |

Data is given for DFT-MD simulations (UKS $M = 1$) of fully hydrated Fe SAC defects with two adsorbed water molecules (*OH₂), with one water and one hydroxide ion adsorbed (*OH⁻) and with an applied electric field: total running time, average Fe spin polarization $s_{Fe}$ and average Fe charges $q_{Fe}$. When two values are indicated, the distribution has two peaks during the simulation time, as can be seen in Supplementary Fig. 4.

previous gas-phase single molecule calculations, except for a quenching of the spin in Fe-pyridine with an adsorbed hydroxide ion, instead of an intermediary spin state with $s_{Fe} \sim 1$.

In the trajectories reported in Table 3, we observe that the hydroxide ion remains continually adsorbed for the whole simulation length (18 ps) in the case of Fe-porphyrin. On the other hand, there are several proton transfers from neighboring water molecules in the case of Fe-pyridine, eventually leading after 7 ps to the recombination of the hydroxide with the hydronium ion (such recombination was also observed for $M = 3$ for Fe-pyridine). The trajectories are given in Supplementary Fig. 7 as well as in Supplementary Movie 3. We follow the proton transfer reaction (here from liquid water to the adsorbed *OH⁻) by computing the coordination number of the adsorbed oxygen atom *O of the hydroxide, using the switch function

$$n_{*O} = \sum^{n_H} \frac{1 - \left(\frac{r_{*OH}}{r_0}\right)^6}{1 - \left(\frac{r_{*OH}}{r_0}\right)^{18}}, \quad (1)$$

where the sum runs over all hydrogens in the simulation box, $r_{*OH}$ is the distance to the adsorbed oxygen and we choose $r_0 = 1.4$ Å. The choice of exponents in Eq. (1) and cutoff value $r_0$ influences the average number of hydrogen atoms accounted for *O but not the shape of the distributions nor its qualitative behavior. In Fig. 3, we plot a transient free energy profile, taken as the negative logarithm of the frequency distributions of $n_{*O}$, $\Delta F_{transient} = -k_B T \ln(\mathcal{P}(n_{*O}))$. For simulations starting from two adsorbed water molecules (see Fig. 3a), the coordination number has a single minimum at 1.75, while for simulations with an initial hydroxide ion (see Fig. 3b) the distributions initially show a minimum at 0.95, as expected. These simulations have large energy barriers. Due to the lack of multiple barrier crossing events, the free energies shown in Fig. 3 (and Fig. 4c) are not converged to equilibrium profiles and thus are transient quantities. Nonetheless, in Fig. 3b, we observe in the case of Fe-porphyrin a single stretched well corresponding to the hydroxide species, while for Fe-pyridine we observe a reprotonation of the anion, leading to a second minimum corresponding to H₂*O. We stress that for Fe-pyridine only three barrier crossings are observed during the simulation, so that the height of the energy barrier is not equilibrated and depends on the trajectory length. These results show that our proposed Fe-porphyrin structure stabilizes the product of water deprotonation, i.e. the hydroxide anion coordinated to the Fe atom, which adopts an intermediary spin state. On the contrary, for the Fe-pyridine configuration the hydroxide ion is only metastable and will eventually accept a

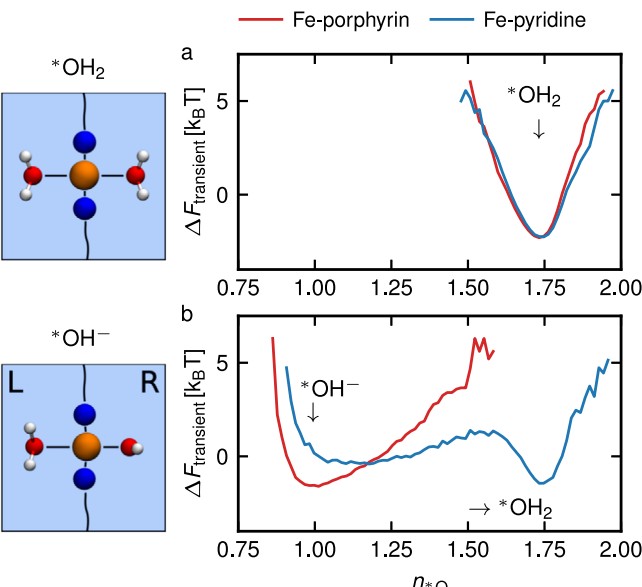

**Fig. 3 Characterization of adsorbed water and hydroxide stability.** Transient free energy profile $\Delta F_{transient}$, taken as the negative logarithm of the frequency distribution of the coordination number of the right adsorbed oxygen atoms $n_{*O}$, defined in Eq. (1), as a function of $n_{*O}$, for simulations with two adsorbed water molecules (**a**) and for simulations with initially a single hydroxide ion adsorbed at the catalytic site (**b**), for the Fe-porphyrin defect (red lines) and the Fe-pyridine defect (blue line). These distributions are taken along the finite length of our simulations (approximately 10–20 ps).

proton from the liquid water so that the adsorbed hydroxide and free hydronium recombine on the 5-10 ps timescale.

**Biasing the deprotonation with E-fields.** To go further in the characterization of the proton transfer, we apply a moderate electric field to trigger the water deprotonation at the surface, which is both a sampling bias method but can also be realized experimentally. We use an external electric field, with a nominal field strength of $D/\varepsilon_0 = 1$ V nm⁻¹, which is small enough to avoid the disruption of the hydrogen bond network. Due to the way electrostatics are treated in periodic boundary conditions in conjunction with polarization of the water in our simulation box the effective electric field is different from the applied one. We evaluate the effective electric field in the bulk water to be $E_{bulk} = 0.12 \pm 0.02$ V nm⁻¹ (see Supplementary Fig. 8 for details). Notably, this electric field stimulus is around 30 times smaller than the one applied in recent DFT-MD simulations of bulk water in order to trigger water dissociation[37,38]. Upon application of the external field, we observe an asymmetry in the Fe–*O distance, shown in Fig. 4b: given that the field is applied in the z direction, the right adsorbed molecule in the $z > 0$ plane aligns strongly with the field and its Fe–*O distance shortens (solid lines), while the left molecule is pulled away from the surface, leading to a lengthening of the bond (dashed lines), with values close to those obtained from the adsorption of a hydroxide anion (see Fig. 2d). This bond shift is pronounced for Fe-porphyrin but weaker for Fe-pyridine. For the Fe-pyridine, we only observe a low spin state with $s_{Fe} \leq 0.5$ and no water deprotonation event occurs over the length of the simulation (around 17 ps), as shown by the distribution of the coordination number of the right water molecule in Fig. 4c (blue line), which displays a single well corresponding to H₂*O. We note that the well is asymmetric and stretched toward the low coordination numbers compared to

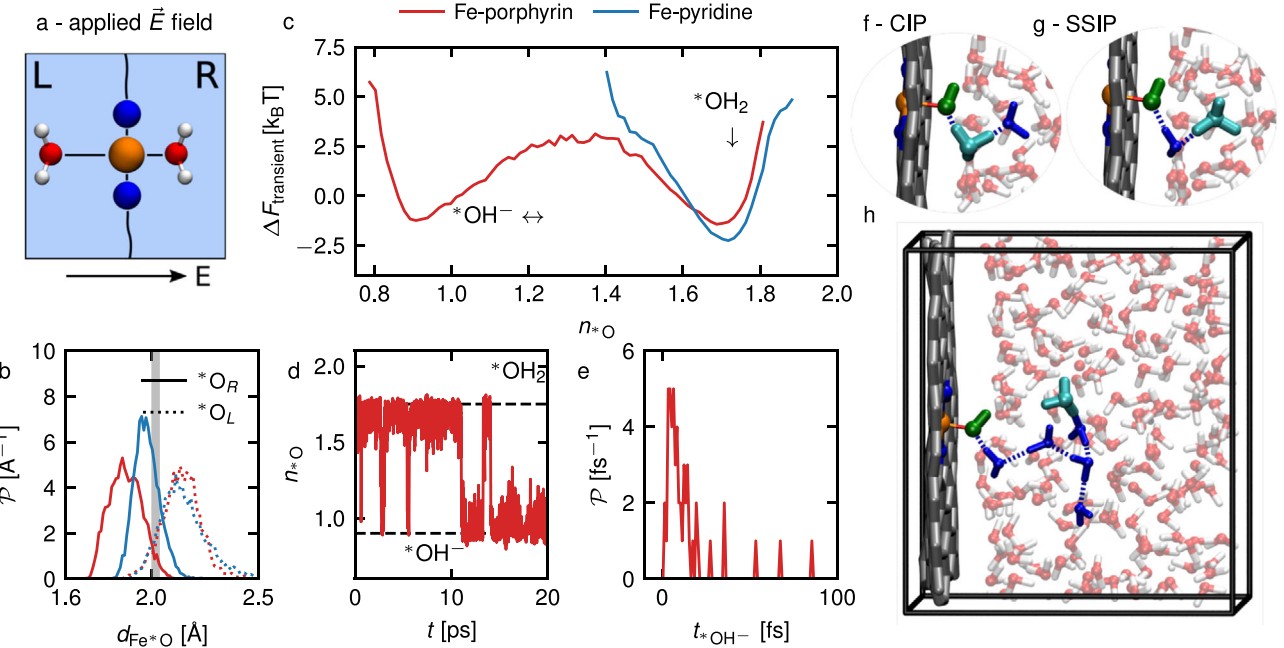

**Fig. 4 Deprotonation under an electric field. a** Schematic representation of fully hydrated simulations with a small applied electric field $D/\varepsilon_0 = 1$ V nm$^{-1}$, with initially two adsorbed water molecules on the catalytic site. We show results for the Fe-porphyrin (red lines) and the Fe-pyridine (blue lines) hydrated defects. **b** Fe–$^*$O distance $d_{Fe^*O}$ distributions for simulations with an applied electric field, for the right adsorbed oxygen (solid lines) and for the left adsorbed oxygen (dotted lines). The vertical gray line shows the average distance for the simulations with two adsorbed water molecules without electric field. **c** Transient free energy profile $\Delta F_{transient}$, taken as the negative logarithm of the probability distribution of the coordination number of adsorbed oxygen atoms $n_{^*O}$, defined in Eq. (1), as a function of $n_{^*O}$. **d** Coordination number of the right adsorbed oxygen $n_{^*O}$ along a Fe-porphyrin trajectory with an applied electric field (red line). Horizontal dashed lines indicate the two maxima in the $n_{^*O}$ distributions, corresponding to the adsorbed water being protonated ($^*OH_2$) or deprotonated ($^*OH^-$). **e** Distribution of the lifetime of the deprotonated species $^*OH^-$ along the previous trajectory. **f–h** Snapshots of the proton transfer at $t = 11$ ps: contact ion pair (**f**), solvent separated ion pair (**g**) and snapshot with separated hydroxide and hydronium ions where we highlighted in blue the water chain through which the proton has hopped (**h**).

Fig. 3a, because of the bias due to the applied electric field. In the case of Fe-porphyrin, we observe low and intermediary spin states with $s_{Fe} = 0.5$ and $0.9$.

Most interestingly, for Fe-porphyrin we observe deprotonation and reprotonation events, as indicated by the double-well structure in Fig. 4c and as can be seen in Supplementary Movie 4. The data is collected along two independent biased simulations of Fe-porphyrin for a total of 35 ps, during which we detect ~120 proton transfer events (including both deprotonation and protonation events). Here and in the following analysis, we define a proton transfer event when a proton's nearest oxygen changes to or from the adsorbed oxygen $^*O$. In these biased simulations, the created proton eventually diffuses away, pushed by the electric field, so that we do not observe any more recombinations between the adsorbed hydroxide and the solvated hydronium ion on the timescale of our simulation. An example of such a trajectory is given in Fig. 4d, where we plot the timeseries of the coordination number $n^*O$ along a 20 ps trajectory. We additionally plot the lifetime distribution of the $^*OH^-$ species in Fig. 4e. We observe several short-lived proton transfers, with a lifetime of less than 25 fs, and a few long-lived transfers of a few hundreds of fs. These short proton transfers are mostly resulting in contact ion pairs (such as in Fig. 4f), or solvent-separated ion pairs (with a single water molecule separating the hydroxide and the hydronium ions, see Fig. 4g). At 11 ps, there is a water deprotonation event that lasts for 2.5 ps, where the created hydronium ion hops relatively far from the adsorbed hydroxide, along an oxygen chain containing 5-6 oxygens, highlighted in blue in Fig. 4h. The hydronium and the hydroxide ions then recombine by traveling backwards through the same water chain,

but another proton transfer occurs after 1 ps, using once again the same water chain, and the hydronium ion diffuses away without returning to the catalytic site for at least 6 ps. As known in the literature[39–41], this highlights the importance of hydrogen bond networks for long-lived water dissociation. In the second independent simulation we observe qualitatively the same behavior.

Finally, using the simulations for Fe-porphyrin with an applied electric field, we analyze the proton transfers occurring at the catalytic site. In Fig. 5, we show four properties as a function of the time difference to a transfer, by plotting the timeseries 300 fs before and after each event and aligning them such that the proton transfer occurs at $t - t_{PT} = 0$. We show averages over all observed deprotonation (green lines) and protonation (purple lines) trajectories. Figure 5 shows, for a proton transfer between the right adsorbed $^*O$ (see Fig. 4a) and a second O from the solvent, the $^*O$–H distance (panel a), the $^*O$–O distance (panel b), the Fe–$^*O$ distance (panel c) and the Fe spin polarization $s_{Fe}$ (panel d). The plots highlight how these properties change during the proton transfer: all trajectories closely overlap, as shown by the small spread of data indicated by the shaded areas, pointing to a common mechanism. Importantly, there is a time-reversal symmetry between protonation and deprotonation trajectories with respect to $t - t_{PT} = 0$, so that there is no time delay in the proton transfer. The $^*O$–H distance increases from 1.05 Å to around 1.6 Å upon deprotonation (and vice-versa for protonation) as expected, while the $^*O$–O distance decreases during the proton transfer, as shown in Fig. 5b. This mechanism was already pointed out in bulk water and it was explained by the fact that the proton transfer barrier is sensitive to the oxygen–oxygen distance, so that shortening the

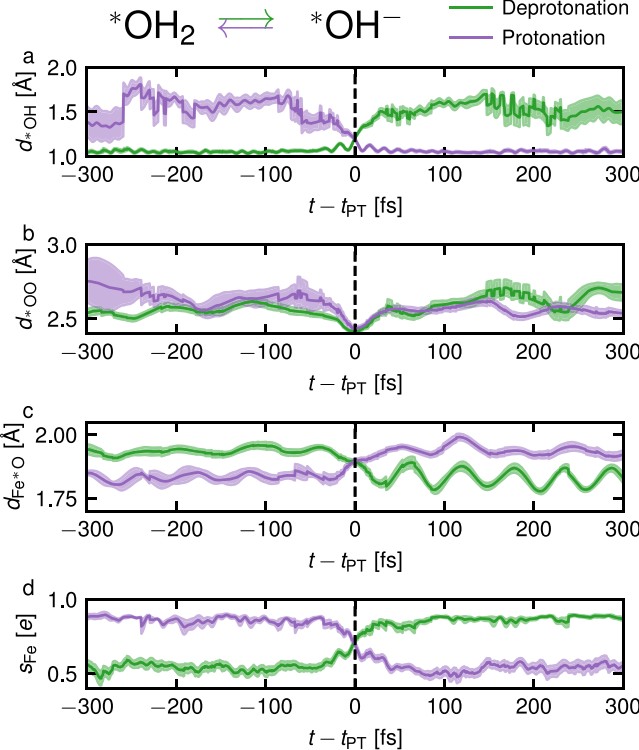

**Fig. 5 Proton transfer dynamics.** *O–H distance $d_{*OH}$ (**a**), *O–O distance $d_{*OO}$ (**b**), Fe–*O distance $d_{Fe*O}$ (**c**) and Fe spin polarization $s_{Fe}$ (**d**) as a function of the time delay to a deprotonation (green line) or a protonation event (purple line), averaged over all proton transfer events (between an adsorbed oxygen *O and a second oxygen O from the solvent) in the simulations of Fe-porphyrin under an applied electric field. Shaded areas correspond to the standard error.

distance favors the proton transfer[39,42,43]. Contrary to the *O–H and *O–O distances, both the Fe–*O distance and the spin polarization $s_{Fe}$ are symmetric with respect to the initial and final states, i.e. they cross $t = t_{PT}$ in the middle between the two minima: for example, $s_{Fe}$ goes from 0.5 to 0.9 during the proton transfer and crosses $t = t_{PT}$ at 0.7.

Finally, for both protonation and deprotonation events, it takes the system around 100 fs to dissipate the perturbation created by the proton transfer: all depicted properties start deviating from their average value around 50 fs before the event and reach their new equilibrium value 50 fs after the event, indicating proton transfer several times faster than what has been reported for bulk water auto-dissociation in the literature, where deviations from the average values up to 600 fs before the proton transfer have been reported[39]. This speed up of reaction time can be interpreted with respect to the catalytic action of the Fe-SAC, combined with the action of the electric field, as well as a difference in hydrogen bonding of the adsorbed water molecule.

## Conclusion

In this work, we have used static DFT and DFT-MD with spin-polarized calculations to thoroughly study the chemical reactivity of two Fe-SAC defects in aqueous conditions in order to mimic an in-situ electrocatalyst. We showed that both defects are reactive in liquid water, adsorbing two interfacial water molecules on both sides. We also highlight a different reactivity of both defects with respect to the water dissociative adsorption process. Indeed, the Fe-porphyrin defect shows a stronger interaction with adsorbed water molecules and a larger stabilization of the product of water dissociation, i.e. of

hydroxide ions, with respect to Fe-pyridine. The Fe-porphyrin defect therefore lowers the energy barrier to water dissociation more than Fe-pyridine. This is confirmed by biased simulations with an applied electric field, in which we observe deprotonation of an adsorbed water molecule only in the case of Fe-porphyrin and not for Fe-pyridine. This points to a higher intrinsic chemical reactivity of the Fe-porphyrin defect with respect to deprotonation. Based on our simulations, we therefore recommend that the SAC synthetic route promotes selectively porphyrin-like defects to catalyze water.

In a more general perspective, very few metal catalysts are prone to water dissociation in aqueous media[44], such as Ru(0001). Even on the prototypical highly reactive Pt(111) catalyst, interfacial water molecules are predicted to remain intact[45]. Such favorable water activation into its self-ions permitted by the Fe-porphyrin SAC defect becomes therefore of great importance to guide the next generation of sustainable electrocatalysts.

## Methods

The two Fe-SAC considered in this study are composed of one iron (Fe) atom coordinated by 4 nitrogens embedded in a single graphene sheet. The geometry of the Fe-porphyrin and Fe-pyridine defects is optimized with density functional theory (DFT) to obtain the lateral dimensions of the simulated box: $19.6896 \times 21.3146$ Å$^2$ for the Fe-pyridine defect and $19.6907 \times 21.0636$ Å$^2$ for the Fe-porphyrin defect. The hydrated Fe-SAC defects contain 201 additional water molecules, with a periodic box length of 17.7206 Å in the direction perpendicular to the defects. All the MD simulations are then run with a CSVR thermostat at 323.15 K, with a timestep of 0.5 fs for at least 10 ps. If not stated otherwise, DFT calculations (static and MD) are performed with the software CP2K[30], using the PBE-D3 functional[31,32] with Grimme's dispersion correction[46] and the DZVP-MOLOPT-SR basis set[47], combined with GTH pseudopotentials[48], and with a cutoff of 600 Ry. Extensive simulation details and in particular for the spin constraints are provided in Supplementary Methods and Notes 1–5.

## Data availability

The data that support the findings of this study are openly available on Zenodo[50] at https://doi.org/10.5281/zenodo.8406389. All other data are available from the corresponding author upon reasonable request.

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

## Acknowledgements

The authors acknowledge support provided by the European Research Council under the European Union's Horizon 2020 research and innovation program (Grant Agreement No. 835117), the Deutsche Forschungsgemeinschaft, Grant No. CRC 1349, Code No. 387284271, Project No. C04 and CRC 1609. We gratefully acknowledge computing resources from the Hochleistungsrechenzentrum Norddeutschland, Project No. bep00106 as well as from the CURTA HPC cluster at ZEDAT, FU Berlin[49]. M.L.B. acknowledges the French HPC resources of GENCI for the grant A0130807364 and funding from the EU H2020 Framework Programme/ERC Synergy Grant agreement number 101071937 n-AQUA.

## Author contributions

M.-L.B. designed the research. L.S and M.R.B. performed the simulations, under the supervision of M.-L.B. and R.R.N. All authors analyzed the simulations and discussed the results. L.S., M.R.B., and M.-L.B wrote the paper with inputs from R.R.N.

## Competing interests

The authors declare no competing interests.
