## [Peer Review File · Communications Chemistry]

Reviewers' comments:

Reviewer #1 (Remarks to the Author):

The paper "Single Atom Catalysis in aqueous conditions: enhanced interfacial water dissociation on a Fe-porphyrin graphene defect" addresses the problem of the activity of single atom catalysts from a theoretical point of view. The authors have performed static and molecular dynamics (MD) DFT calculations on the structure of two variants of a Fe atom incorporated in N-doped graphene. There is a huge computational activity on SACs, but often this is based on screening large numbers of structures with several crude approximations. This is not the case of this paper where the nature of a specific SAC is carefully addressed with particular attention to the role of water. The authors have used the correct approach and performed MD simulations with large supercells and sufficiently long simulation times to detect when water dissociation occurs on the two variants of the catalyst. The work done is interesting and novel and deserves to be published.

However, there are some aspects that need to be addressed before publication is recommended. The authors are right when they say that spin-polarization is essential in the study of Fe-based SACs and in fact investigate high and low spin states. For the MD simulations they assume a singlet ground state. However, the entire work is based on the semi-local PBE functional. This functional is not well suited to describe localized spin states, as it is the case for a transition metal embedded in a support. It would have been good to check the PBE results against at least PBE+U or even better hybrid functional calculations. The differences in adsorption energies changing functional can be substantial (see e.g. A. M. Patel, S. Ringe, S. Siahrostami, M. Bajdich, J. K. Nørskov, A. R. Kulkarni, *The Journal of Physical Chemistry C* 2018, 122, 29307; these authors concluded that "the blind use of GGA functionals to describe single-atom catalysts may produce inaccurate results"). This is also the experience of the referee.

Maybe this is not the case for water, but it should be shown. I am not suggesting to repeat the entire work at the PBE+U level, but at least a few checks are in order, and in particular:

- The difference in energy between $M = 1$ and $M = 3$ electronic configurations should be checked at the PBE+U or PBE0 levels (or both) and compared with PBE
- The same should be done for the adsorption energy of water and the OH- fragment. Single point calculations on PBE-optimal structures would be sufficient.

Minor change: on p. 3 it is mentioned that "a hydroxide ion is adsorbed, as in Fig. 1G". I assume that this occurs simultaneously to the addition of a proton in solution, in order to keep electroneutrality. In this case, it should be specified.

Reviewer #2 (Remarks to the Author):

The authors used spin-polarized ab initio molecular dynamics to thoroughly investigate the

chemical reactivity of two Fe-SAC defects in aqueous conditions to mimic an in situ electrocatalyst. There are still the following issues that need to be addressed:

1. Is the Fe-porphyrin graphene defect the predominant configuration in Fe deposited in N-doped graphite materials? Is there any corresponding literature to support it?
2. Are these two Fe-N4 SAC defects experimentally easy to synthesize? What is the guiding significance for the synthesis of the catalysts?
3. Generally, the defects could change the density of electronic states of carbon atoms at the defects in the graphene layer to form active sites. What is the difference in the density of electronic states between the two Fe-SAC defects?

Reviewer #3 (Remarks to the Author):

The aim of the manuscript was to investigate the interfacial water dissociation properties of Fe-porphyrin graphene defects in aqueous conditions. The computed results demonstrated that two interfacial water molecules are spontaneously adsorbed by Fe Single Atom Catalysts (SACs) on opposite sides. While the results may appear reasonable, the authors did not explain the following issues:

1. The coaxial adsorption of two water molecules occurs step by step, rather than simultaneously, in two opposite sites. From a binding energy standpoint, the adsorption of the second water molecule is expected to be more challenging compared to the adsorption of the first water molecule. Regrettably, the authors disregarded this aspect in their analysis.
2. On Page 2, I came across the statement "In previous studies, the simulation box was only slightly larger than the catalytic center itself." The choice of the calculation model is based on the software utilized and the underlying theoretical calculation algorithms. For instance, when employing VASP software for the calculation of long-range ordered materials, only one crystal cell is used as the calculation model, and the software automatically incorporates the impact of material periodicity on the calculation model. Hence, I am inclined to believe that selecting a larger simulation box is not necessarily superior to choosing a smaller but more appropriate one.
3. As is widely known, it is very challenging for 2D materials to exist in a single-layer form, even when solvents are present. However, it appears that the authors utilized a monolayer catalyst model for calculations in the manuscript.

Perhaps the authors possess a deeper understanding of these scientific issues, so I recommend that they provide more clear explanations of them in the manuscript. By doing so, this manuscript can be accepted by <Communications Chemistry>.

Revised Manuscript

Single Atom Catalysis in aqueous conditions: enhanced interfacial water dissociation on a Fe-porphyrin graphene defect

Laura Scalfi, Maximilian R. Becker, Roland R. Netz, and Marie-Laure Bocquet

E-mail:

General remarks

We thank the reviewers for their evaluation of our work. Taking into account their comments and questions, we have added extensive calculations to the manuscript, in particular ligand adsorption energies on the molecular complex and a thorough comparison with hybrid exchange-correlation functionals (PBE vs PBE0). With these additions, we have decided to remove the RKS data (no spin) from the main text, since it is clear that enabling spin states in the catalyst is crucial. A detailed response to the comments and issues raised by each reviewer is provided in the following in blue. Changes made to the main text or the supplemental information are described in red.

Response to Reviewer #1

Reviewer comment:

The paper "Single Atom Catalysis in aqueous conditions: enhanced interfacial water dissociation on a Fe-porphyrin graphene defect" addresses the problem of the activity of single atom catalysts from a theoretical point of view. The authors have performed static and molecular dynamics (MD) DFT calculations on the structure of two variants of a Fe atom incorporated in N-doped graphene. There is a huge computational activity on SACs, but often this is based on screening large numbers of structures with several crude approximations. This is not the case of this paper where the nature of a specific SAC is carefully addressed with particular attention to the role of water. The authors have used the correct approach and performed MD simulations with large supercells and sufficiently long simulation times to detect when water dissociation occurs on the two variants of the catalyst. The work done is interesting and novel and deserves to be published. However, there are some aspects that need to be addressed before publication is recommended.

We thank the reviewer for their evaluation of our work and suggestions to improve it.

Reviewer comment:

*The authors are right when they say that spin-polarization is essential in the study of Fe-based SACs and in fact investigate high and low spin states. For the MD simulations they assume a singlet ground state. However, the entire work is based on the semi-local PBE functional. This functional is not well suited to describe localized spin states, as it is the case for a transition metal embedded in a support. It would have been good to check the PBE results against at least PBE+U or even better hybrid functional calculations. The differences in adsorption energies changing functional can be substantial (see e.g. A. M. Patel, S. Ringe, S. Siahrostami, M. Bajdich, J. K. Nørskov, A. R. Kulkarni, *The Journal of Physical Chemistry C* 2018, 122, 29307; these authors concluded that “the blind use of GGA functionals to describe single-atom catalysts may produce inaccurate results”). This is also the experience of the referee. May be this is not the case for water, but it should be shown. I am not suggesting to repeat the entire work at the PBE+U level, but at least a few checks are in order, and in particular:*

- The difference in energy between $M = 1$ and $M = 3$ electronic configurations should be checked at the PBE+U or PBE0 levels (or both) and compared with PBE*
- The same should be done for the adsorption energy of water and the OH- fragment. Single point calculations on PBE-optimal structures would be sufficient.*

We thank the reviewer for pointing out this issue. As suggested, we studied the adsorption energy of two water molecules or of a hydroxide ion and a water molecule on the Fe-defects comparing PBE and PBE0 functionals. The comparison is shown in Fig. 1A: for most gas phase configurations with water molecules, the open circles lie close to $x=y$ curve i.e. the absolute difference between PBE and PBE0 results is within 2.5 kcal/mol – as was also observed for H₂O adsorption by Patel et al (JPCC 2018, 122, 29307 - the authors stated "As expected, we predict similar *H₂O binding energies using different methods."). For hydroxide adsorption energies, the crosses deviate below the $x=y$ curve, i.e. the difference is larger, with energies ~ 10 kcal/mol more negative for PBE than PBE0. We note that these calculations are difficult to perform due to different local minima corresponding to different spin states, which render the calculations sensible to the choice of initial conditions. The stability of OH⁻ on the defects in solution might therefore be overestimated using the classical PBE functional, but the energy difference between PBE and PBE0 results remains roughly constant, i.e. the differences observed between Fe-pyridine and Fe-porphyrin are likely to be robust.

We also evaluated the difference in energy between system multiplicities $M = 1$ and $M = 3$, noted $\Delta E = E_{M3} - E_{M1}$, for PBE and PBE0. For this, we used selected gas phase configurations with either two water molecules adsorbed onto the defect or a water molecule and a hydroxide ion (which were geometry-optimized on the PBE

Figure 1: Comparison of PBE (y-axis) with PBE0 (x-axis), for a selection of gas-phase configurations (presented in Table II of the main text) for the Fe-pyridine defect (blue circles) and the Fe-porphyrin defect (red circles). (A) BSSE-corrected adsorption energies of either two water molecules or a hydroxide ion and a water molecule, obtained from single point calculations. (B) Energy differences between system multiplicities $M = 3$ and $M = 1$, evaluated from two single point calculations of the same configuration.

level), for which we ran single point energy calculations. The results are shown in Fig. 1B. We find that all points lie close to the diagonal, i.e. the energy differences between system multiplicity $M = 1$ and $M = 3$ agree rather well between PBE and PBE0, indicating that the use of PBE is also suitable for our case.

To further assess the use of PBE in the liquid phase, we additionally computed ΔE for a set of configurations along the MD simulations with either only liquid water (H_2^*O) or with an additional hydroxide anion adsorbed on the Fe-defect ($^*\text{OH}^-$) and a hydronium cation in the bulk to compensate the charge. The results are shown in Fig. 2. As expected, the data lies mostly on the left of the $x = y$ curve, i.e. PBE0 favors the higher spin state with respect to PBE. For the vast majority of the configurations, PBE predicts that $M = 1$ configurations are energetically more stable than $M = 3$ configurations, which is why we chose to discuss simulations with system multiplicity $M = 1$ in the main text. For Fe-pyridine configurations, PBE0 predicts that the system multiplicity $M = 1$ is lower in energy for most configurations studied. For Fe-porphyrin, although we do observe that there is a fraction of the configurations – mostly with liquid water – for which PBE0 predicts that $M = 3$ is more stable, we find a significant fraction of configurations for which $M = 1$ is more stable than $M = 3$, especially for configurations

Figure 2: Comparison of the energy difference between system multiplicities $M = 3$ and $M = 1$ obtained from PBE with respect to PBE0, for a selection of liquid-phase configurations taken along the DFT-MD trajectories (see Table III in the main text). We distinguish configurations including a hydroxide ion (crosses) or only water molecules (circles), for the Fe-pyridine defect (blue/cyan symbols) and the Fe-porphyrin defect (red/orange symbols). The data lies mostly in the $y > 0$ region (shaded area), where PBE predicts the $M = 1$ multiplicity to be more stable than $M = 3$. The darker green area corresponds to the region where also PBE0 predicts $M = 1$ to be more stable.

with a hydroxide ion. We therefore confirm our choice of a system multiplicity $M = 1$ to run MD simulations for Fe-pyridine systems, while precautions should be taken for Fe-porphyrin.

We included this discussion as well as both figures in a new section in the SI. We also added a short discussion in the main text:

"Because of the localized d -orbitals on the isolated Fe metal ion, the delocalization error of GGA (generalized gradient approximation) functionals might lead to inaccurate results, while hybrid functionals such as PBE0 correct for this and are therefore more reliable for predicting properties of the transition metal ions present in SACs. However, this is prohibitively computationally expensive for running extensive MD simulations. To check for the severity of this approximation, we provide a comparison of adsorption energies of single ligands onto the defects and of the energy difference between spin states calculated with PBE and PBE0. The data is given in Fig. S5 and S6 in the SI and discussed in sections II.B and III."

Although this was not clear enough in our previous version, we did run both $M = 1$ and $M = 3$ MD simulations

to check the differences between them (shown in SI).

We now made this point clearer in the main text and refer to the results for different spin multiplicities in the SI:

"For completeness, we run both UKS simulations with a system multiplicity $M = 1$ and $M = 3$. For a selection of configurations along these DFT-MD trajectories, we calculate the difference between these two settings and found the $M = 1$ setting to be the lowest in energy in the vast majority of the cases (see Fig. S6 in the SI, where we also give a comparison with PBE0 results). Therefore, in the following, we only show results of extensive DFT-MD simulations using a system multiplicity $M = 1$. Other spin settings in DFT-MD runs are analyzed in Tab. S1 and Fig. S4 in the SI."

Minor change: on p. 3 it is mentioned that "a hydroxide ion is adsorbed, as in Fig. 1G". I assume that this occurs simultaneously to the addition of a proton in solution, in order to keep electroneutrality. In this case, it should be specified.

For the simulations in solution, we indeed add a proton to compensate the negative charge of the hydroxide ligand (as specified on page 6). However for the gas phase configurations, we did differently in our previous version and considered systems with a hydroxide ion holding a net charge of -1. During this revision, we have checked that using charged systems or neutral systems (adding a single proton H^+ far from the Fe-defect as a counterion) does not impact the geometry, charges or spin polarizations obtained after geometry optimization in the gas phase. Finally, because we decided to include adsorption energies in both gaseous phase and liquid phase, we only consider neutral systems with a counterion when needed for sake of consistency.

We have updated Table II of the main text and clarified this point.

"In the case of $*OH^-$, a single proton is added far from the hydroxide ion to ensure the neutrality of the simulation cell. Note that to ensure a +1 charge on the proton, no basis set is attributed to the H atom."

Response to Reviewer #2

Reviewer comment:

The authors used spin-polarized ab initio molecular dynamics to thoroughly investigate the chemical reactivity of two Fe-SAC defects in aqueous conditions to mimic an in situ electrocatalyst. There are still the following issues that need to be addressed:

We thank the reviewer for their evaluation of our work and suggestions to improve it.

1. Is the Fe-porphyrin graphene defect the predominant configuration in Fe deposited in N-doped graphite materials? Is there any corresponding literature to support it?

Multiple experimental studies (Li et al, ACS Catalysis 8, 8450–8458, 2018/ Fei et al, Nature Communications 6, 8668, 2015/ Zitolo et al, Nature Materials 2015 14:9 14, 937–942, 2015/ Yang et al, PNAS 115, 6626–6631, 2018) have found, using XANES measurements in conjunction with DFT data or XPS, that both the pyridine and porphyrin motifs occur in the synthesized catalysts, roughly in similar proportions. The exact relative density of the two motifs depends on the exact synthesis pathway of the material and is usually not provided explicitly. Recently however, Zhang *et al.* (Energy & Environmental Science 13, 111–118 (2020)) have reported that, designing a suitable synthesis pathway, the relative proportion of Fe-porphyrin and Fe-pyridine sites can be tuned, up to nearly entire purification of the Fe-porphyrin motif.

To answer this point and the next, we have considerably rewritten the introduction part and added a more detailed discussion of the vast literature on this point.

2. Are these two Fe-N4 SAC defects experimentally easy to synthesize? What is the guiding significance for the synthesis of the catalysts?

It is a very interesting question and we have now added and summarized various synthesis strategies in the introduction part. To our understanding, three routes are developed that we now explicit in the introduction.

3. Generally, the defects could change the density of electronic states of carbon atoms at the defects in the graphene layer to form active sites. What is the difference in the density of electronic states between the two Fe-SAC defects?

As suggested by the reviewer, we calculated densities of states projected on selected carbon atoms that present a modification of their environment with respect to a pure graphene sheet (see Fig. 3). Indeed, Yang et al. (PNAS 115.26 (2018): 6626-6631) indicate that for the oxygen reduction reaction in case of a Fe-porphyrin defect eight of the carbon atoms become catalytically active sites, while the same does not happen for the Fe-pyridine defect. They argue this is due to a shift of the p-orbitals band of Fe-pyridine carbons towards lower energies (~ 3 eV)

with respect to the Fe-porphyrin ones. In our calculations, we only observe a very small energy downshift (roughly 0.7 eV) in the pDOS of carbons bound to nitrogen between the Fe-pyridine as compared to those from the Fe-porphyrin defects. The trend is hence similar albeit not enough conclusive for us. Also, in our DFT-MD studies, we do not find an increased activity of carbon atoms, *i.e.* during our MD simulations, no bonding to other nitrogen or carbon atoms is observed.

Figure 3: Projected Density of States (pDOS) onto the s- and p-shell of the C atoms bonded to a nitrogen atom for the Fe-porphyrin (A) and for the Fe-pyridine SAC defect (B). The energy is given with respect to the Fermi level and the vertical dashed red line indicates the average position of the p-orbital band.

Response to Reviewer #3

Reviewer comment:

The aim of the manuscript was to investigate the interfacial water dissociation properties of Fe-porphyrin graphene defects in aqueous conditions. The computed results demonstrated that two interfacial water molecules are spontaneously adsorbed by Fe Single Atom Catalysts (SACs) on opposite sides. While the results may appear reasonable, the authors did not explain the following issues:

We thank the reviewer for their evaluation of our work and suggestions to improve it.

1. *The coaxial adsorption of two water molecules occurs step by step, rather than simultaneously, in two opposite sites. From a binding energy standpoint, the adsorption of the second water molecule is expected to be more challenging compared to the adsorption of the first water molecule. Regrettably, the authors disregarded this aspect in their analysis.*

We respectfully disagree with the referee's statement that the adsorption of water molecules do occur step by step. In contrast it occurs simultaneously. Our MD simulations with Fe-graphene defects in explicit liquid water were setup starting from an equilibrium configuration of water solvent, where no water molecule is adsorbed at the surface. Fig. 4 shows the evolution of the Fe-O distance of the two closest waters on either side of the sheet when put in contact with the Fe-porphyrin defect (triplet state). We observe a fast simultaneous adsorption of both molecules within 500 fs.

Figure 4: Fe-O distance between the Fe atom and the two closest water molecules during the equilibration, in the case of the Fe-porphyrin system.

However following the reviewer's (and also the first reviewer) interest about binding or adsorption energies that we indeed disregard in our previous version, we now add new data showing that water molecules and the hydroxide ion adsorb more strongly on Fe-porphyrin than on Fe-pyridine, which is in agreement with our study. Moreover,

the adsorption energies of H_2O and especially OH^- onto the Fe-defect are rather large, indicating how strong the interaction with the Fe-defect is.

We added the new adsorption data to Table II of our manuscript, along with the following discussion:

"For each system, we compute adsorption energies of the water molecules and/or the hydroxide ion onto the Fe-defect. The adsorption energy of two water molecules is about -20 kcal/mol, while we observe a very strong adsorption of OH^- onto both defects with a total adsorption energy between -110 to -135 kcal/mol. Interestingly, ΔE_{ads} is globally more negative for Fe-porphyrin configurations than for Fe-pyridine ones, *i.e.* water molecules and the hydroxide ion adsorb more strongly on Fe-porphyrin than on Fe-pyridine. Values obtained from PBE0, given in Fig. S5 the SI, are in qualitative agreement with the conclusions above. The relative adsorption energies between Fe-pyridine and Fe-porphyrin are similar, nevertheless we find that ΔE_{ads} is more negative for PBE, with differences of about 2.5 kcal/mol for adsorption of water molecules and 10 kcal/mol for adsorption of OH^- ."

2. On Page 2, I came across the statement "In previous studies, the simulation box was only slightly larger than the catalytic center itself." The choice of the calculation model is based on the software utilized and the underlying theoretical calculation algorithms. For instance, when employing VASP software for the calculation of long-range ordered materials, only one crystal cell is used as the calculation model, and the software automatically incorporates the impact of material periodicity on the calculation model. Hence, I am inclined to believe that selecting a larger simulation box is not necessarily superior to choosing a smaller but more appropriate one.

We agree with the reviewer that, when the system studied is a periodic crystal, studying a single unit cell is preferred. However, in this work we focus on Fe-defects embedded in a graphene support : the Fe-N₄ defects do not constitute a periodically replicated unit cell, but they are more likely "diluted" and randomly scattered on the graphene support: experimentally, the reported density of defects is between 0.5 and 4 wt% (see the review by Zhang et al, Small Methods 3, 1800443 (2019)). Our setup corresponds to 2.8-2.9 wt%, consistent with this range. However, periodically replicating a small unit cell such as a single porphyrin structure is not a faithful representation of the system. This has been done in the literature, but the structures obtained, shown schematically on Fig. 5, correspond to ~ 16 wt%, present strained 6-carbon rings and an 8-carbon ring, and the graphene support is not present anymore. In particular, the disruption of the aromaticity might have serious consequences on the calculated results. Embedding a single Fe-pyridine defect in a large graphene sheet was already done in the literature because the defect is commensurate with the graphene structure. For the Fe-porphyrin defect however, because the structure is not commensurate with graphene, such small cell were used previously.

Most importantly however, our study is not only a gas phase simulation, but we study the liquid phase in

Figure 5: Sketch of a periodically replicated porphyrin.

contact with the defects by MD simulations. To treat liquid phases correctly, it is important that the box sizes are large enough to avoid correlations, *i.e.* the box should be larger than the correlation length of the liquid. This is challenging in DFT because of the numerical cost, but imposes nevertheless a minimum size to our boxes (about 1.5-2 nm).

We have added the experimental densities of defects that justify the use of larger box sizes in the manuscript: "Indeed, the reported experimental density of these defects is between 0.5 and 4 wt%. Therefore, we argue here that larger unit cells should be used to ensure a better representation of the systems and that the local defect created by the SAC insertion induces only small local perturbations".

3. *As is widely known, it is very challenging for 2D materials to exist in a single-layer form, even when solvents are present. However, it appears that the authors utilized a monolayer catalyst model for calculations in the manuscript. Perhaps the authors possess a deeper understanding of these scientific issues, so I recommend that they provide more clear explanations of them in the manuscript. By doing so, this manuscript can be accepted by .*

We agree with the referee that using a monolayer catalyst model is a strong hypothesis and indeed in water the doped graphene sheet is mostly supported on a solid, so that water is exposed on one side. Hence our model captures only partly the local realistic environment. However this configuration is frequently adopted in the vast literature and is also computationally advantageous to simulate.

We have added in the beginning of the "SAC in liquid water" part the following sentence
"Although we are aware that in many experiments the doped graphene sheet is often exposed to water on one side and to a solid support on the other side, we choose this model because this configuration is frequently adopted in the literature and is also computationally advantageous to simulate."

REVIEWERS' COMMENTS:

Reviewer #1 (Remarks to the Author):

The authors have addressed extensively the points raised in the original report, and the paper is now ready to be accepted

Reviewer #2 (Remarks to the Author):

The authors have revised the manuscript according the comments and suggestions from Reviewers. It could be accepted for publication.

Reviewer #3 (Remarks to the Author):

The revised version can be accepted.